# Metabolite Changes in Indonesian *Tempe* Production from Raw Soybeans to Over-Fermented *Tempe*

**DOI:** 10.3390/metabo13020300

**Published:** 2023-02-17

**Authors:** Mahensa Billqys Nurhayati Prativi, Dea Indriani Astuti, Sastia Prama Putri, Walter A. Laviña, Eiichiro Fukusaki, Pingkan Aditiawati

**Affiliations:** 1School of Life Sciences and Technology, Institut Teknologi Bandung, Jalan Ganesha No. 10, Bandung 40132, Indonesia; 2Research and Development Agency, West Java Provincial Government, Jalan Kawaluyaan Indah Raya No. 6, Bandung 40286, Indonesia; 3Department of Biotechnology, Graduate School of Engineering, Osaka University, 2-1 Yamadaoka, Suita 565-0871, Japan; 4Industrial Biotechnology Initiative Division, Institute for Open and Transdisciplinary Research Initiatives, Osaka University, Suita 565-0871, Japan; 5Microbiology Division, Institute of Biological Sciences, University of the Philippines Los Baños, Los Baños 4031, Laguna, Philippines; 6Osaka University-Shimadzu Omics Innovation Research Laboratories, Osaka University, Suita 565-0871, Japan

**Keywords:** *tempe*, soybean soaking, fungal fermentation, metabolite, sugar, amino acid

## Abstract

*Tempe* is fermented soybean from Java, Indonesia, that can serve as a functional food due to its high nutritional content and positive impact on health. Although the *tempe* fermentation process is known to affect its nutrient content, changes in the metabolite profile during *tempe* production have not been comprehensively examined. Thus, this research applied a metabolomics approach to investigate the metabolite profile in each step of *tempe* production, from soybean soaking to over-fermentation. Fourteen samples of raw soybeans, i.e., soaked soybeans (24 h), steamed soybeans, fungal fermented soybeans, and over-fermented soybeans (up to 72 h), were collected. Untargeted metabolomics by gas chromatography/mass spectrometry (GC–MS) was used to determine soybean transformations from various fermentation times and identify disparity-related metabolites. The results showed that soybeans samples clustered together on the basis of the different fermentation steps. The results also showed that sugar, sugar alcohol, organic acids, and amino acids, as well as fermentation time, contributed to the soybean metabolite profile transformations. During the fermentation of *tempe*, sugars and sugar alcohols accumulated at the beginning of the process before gradually decreasing as fermentation progressed. Specifically, at the beginning of the fermentation, gentiobiose, galactinol, and glucarate were accumulated, and several metabolites such as glutamine, 4-hydroxyphenylacetic acid, and homocysteine increased along with the progression of fermentation. In addition, notable isoflavones daidzein and genistein increased from 24 h of fermentation until 72 h. This is the first report that provides a complete description of the metabolic profile of the *tempe* production from soybean soaking to over-fermentation. Through this study, the dynamic changes at each step of *tempe* production were revealed. This information can be beneficial to the *tempe* industry for the improvement of product quality based on metabolite profiling.

## 1. Introduction

*Tempe* is a traditional fermented soybean food that originated from Central Java, Indonesia, centuries ago [1]. From Central Java, *tempe* has spread to many places in Indonesia; thus, it is considered as an Indonesian heritage food. The popularity of *tempe* has spread not only in Asia but also globally. Today, people are becoming more aware of their health and avoiding unhealthy foods such as animal protein sources. Health-conscious people turn to plant sources, including various legumes and their derivatives, such as nonanimal fermentation foods, e.g., *tahu* (tofu) and *tempe*. *Tempe* can be a good protein substitute to meat [2] since it is more economical than meat, easier to digest, and nutritious, as it contains the complete essential amino acids and vitamin B_12_ [3]. More importantly, *tempe* is a known functional food because it contains antioxidants [4] and antimicrobials [5]. For example, it contains isoflavones, such as genistein, daidzein, glycitein, and factor-2 (6,7,4′-trihydroxyisoflavone), an antioxidant known to prevent free radicals that triggers cancer [4]. *Tempe* may also inhibit degenerative diseases, namely, diabetes, cardiovascular disease, cancer risk, high cholesterol, and osteoporosis, as well as relieve menopause symptoms [1,6]. Consequently, more people, including the vegetarian/vegan community, are switching from meat to *tempe*. 

Since *tempe* is made from soybeans, it also contains antinutritional factors, such as trypsin inhibitors, oligosaccharides (sucrose, raffinose, stachyose, melibiose), phytic acid, polyphenol, lectin, and others, which can be removed by fermentation [7,8,9]. The whole process of making *tempe* from soybeans can reduce antinutritional factors, such as oligosaccharides and phytic acid, by up to 33% and improve nutrition and flavor [7,8]. Some compounds, such as vitamin B_12_ [10], antioxidants, and complete essential amino acids, increase along with fermentation time [10,11,12]. 

Generally, *tempe* is manufactured traditionally and not aseptically [13], based on knowledge passed down from generation to generation. There are many different procedures to create *tempe* in different regions of Indonesia; however, in general, it is made by soaking, dehulling, cooking, and inoculating with *Rhizopus* sp. [14,15,16]. According to Kadar et al. (2018), there are at least four methods for processing *tempe*, each yielding different flavor characteristics and appearances. The various processes from different regions also give different metabolite profiles [11]. West Javanese *tempe* has more sugar compared to *tempe* in Central or East Java, which contain more amino acids [11].

The taste characteristics of fermented foods, including sake, miso, soy sauce, and *tempe*, are derived from the free amino acids released from protein during fermentation [17]. Amino acids contribute to the taste characteristics of food, and can provide sweet (glycine, alanine, serine, threonine, and lysine), bitter (isoleucine, leucine, methionine, phenylalanine, valine, arginine, histidine, tryptophan, and tyrosine), sour (aspartate, asparagine, and glutamic acid), umami (aspartate and glutamic acid), and/or meat-like (glutamic acid, cysteine, and methionine) flavors [18]. The soybean fermentation process using *Rhizopus* sp. produces *tempe* with complete essential and nonessential amino acids [11,12,19]. According to previous studies, most if not all proteogenic amino acids can be found in *tempe*. Glutamic acid and alanine are the most abundant amino acids in *tempe* [12]. Alanine gives a sweet taste, while glutamate gives a meaty or umami taste [18].

This study used a nontargeted gas chromatography/mass spectrometry (GC–MS) metabolomics analysis to describe how metabolites change overtime during the *tempe* production process. It is important to know the period when *tempe* has the highest nutritional content, as well as determine whether it contains metabolites that can affect health or harmful compounds. The fermentation process can be stopped when *tempe* has the highest nutritional content and does not contain harmful metabolites. Hence, it is important to monitor the metabolic changes in the whole process to determine whether the fermentation process should be stopped or extended. This study also examined the presence of several metabolites that affect aroma and flavor as a function of the content of amino acids and sugars. Although there have been many publications on *tempe*, this is the first report that provides a complete overview of the metabolic changes in *tempe* production in Indonesia. This can potentially guide the *tempe* production industry on the basis of nutrition, flavor, aroma, and food safety. 

## 2. Materials and Methods

In this research, soybean from KOPTI, The Indonesian *Tempe* and Tofu Producers Cooperative, was used as a raw material for *tempe* production, which was inoculated with Raprima *tempe* starter containing *Rhizopus oligosporus* NRRL 2771 produced by Aneka Fermentasi Industri (AFI) in Bandung, West Java, Indonesia. The general steps in *tempe* production consists of washing the beans, soybean soaking, peeling of the outer skin, cooking of the beans (boiling or steaming), inoculation with culture starter, and packing with perforated plastic, followed by 48 h of fungal fermentation (Figure 1). The *tempe*-making process was conducted in Bandung, Indonesia. During the soaking process, sampling was performed every 6 h, and five samples from 0 h, 6 h, 12 h, 18 h, and 24 h of soaking time were collected. During the FF process, sampling was carried out every 12 h. There were six samples from 0 h, 12 h, 24 h, 36 h, and 48 h and two samples from the OF process at 60 h and 72 h. 

### 2.1. Sample Preparation before Transportation

Each sample was kept in a different tube and stored at −20 °C until all the processes were completed. All the samples were quenched together in liquid nitrogen for 5 min and lyophilized before sending to Japan for metabolomic analysis. Three datasets were used to analyze the difference between the three treatment groups. Dataset 1 was used to observe all the metabolite profiles in the *tempe*-making process, from RS to OF. Dataset 2 was used to investigate metabolite transformations during SS. Dataset 3 was used to examine how metabolites changed in the FF process only.

Previous research noted that tempe samples were sticky during the milling process because of their moisture content. Thus, before grinding, the samples were lyophilized to eliminate the water content. Another issue was that RS and SS0 had extra hard grains compared to the other samples. Therefore, only a small amount of completely dry soybean was needed for the milling process.

### 2.2. Extraction and Derivatization

Metabolite extraction was performed by following published extraction methods with some modifications [11]. In this study, *tempe* samples were lyophilized before and after milling. Lyophilized samples (soybean and *tempe*) were put into a 50 mL tube (Yasui Kikai Co., Osaka, Japan) with a metal cone (Yasui Kikai Co., Osaka, Japan) in it and frozen using liquid nitrogen, before being ground into a fine powder at 2000 rpm for 30–60 s using a multi-bead shocker (Yasui Kikai Co., Osaka, Japan). Soybean/*tempe* powder (15 mg) was extracted using the same methods [11] until hydrophilic and hydrophobic fractions were separated, except that the volume of the upper aqueous phase was adjusted to 200 µL.

Metabolite extraction was performed by following published extraction methods with some modifications [11]. In this study, lyophilized samples (soybean and *tempe*) were put into a 50 mL tube (Yasui Kikai Co., Osaka, Japan) with a metal cone (Yasui Kikai Co., Osaka, Japan) and frozen using liquid nitrogen, before being ground into a fine powder at 2000 rpm for 30–60 s using a multi-bead shocker (Yasui Kikai Co., Osaka, Japan). Soybean/*tempe* powder was kept at −30 °C until the extraction step. A blank for manual annotation was prepared following the same steps as the sample preparation. Soybean/*tempe* powder (15 mg) was transferred to a 2 mL tube and extracted using a single-phase mixed solvent of methanol, ultrapure water, and chloroform in a 5:2:2 ratio, respectively (containing ribitol 0.2 mg/mL as an internal standard) (Wako Pure Chemical Industries, Ltd., Osaka, Japan). The mixture was vortexed for 1 min and then centrifuged at 4 °C and 10,000× *g* for 3 min. The supernatant (900 μL) was transferred into a 1.5 mL tube, and 400 μL of ultrapure water was added to the tube. The mixture was vortexed for 1 min and centrifuged at 4 °C and 10,000× *g* for 3 min; then, hydrophilic and hydrophobic fractions were separated. A 200 µL aliquot of the upper aqueous phase was transferred into a new 1.5 mL tube with a pierced cap. Another 200 µL of all the sample replicates (n = 3) and blank samples were collected and mixed into one pool as quality controls (QCs). The solvent, including QCs, was evaporated by vacuum centrifugation for 1 h using a centrifuge concentrator at room temperature and lyophilized overnight. Derivatization of the metabolite extract by oximation and trimethylsilylation was performed for all the samples, pooled QCs, and the blank at the same time before GC–MS analysis, following the protocol described previously. Oximation was conducted by adding 100 µL of methoxyamine hydrochloride (Sigma-Aldrich, Milwaukee, WI, USA) (20 mg/mL in pyridine (Wako Pure Chemical Industries, Ltd., Osaka, Japan)) to the lyophilized extract, followed by incubation at 30 °C and 1200 rpm for 90 min. Silylation was conducted after oximation by adding 50 µL of *N*-methyl-*N*-trimethylsilyl-trifluoroacetamide (MSTFA) (GL Science, Inc., Tokyo, Japan). The mixture was incubated again at 37 °C and 1200 rpm for 30 min.

### 2.3. GC–MS Analysis 

GC–MS analysis was performed immediately after the derivatization reactions using a GCMSQP2010 Ultra (Shimadzu, Kyoto, Japan) equipped with a 30 m × 0.25 mm i.d. fused silica capillary column coated with 0.25 µm InertCap 5MS/NP (GL Science, Inc.) and an AOC-20i/s (Shimadzu) as an autosampler. System control and data acquisition were conducted using the GC–MS solutions software (Shimadzu). The derivatized samples (1 µL) were injected in split mode (12:1 (*v*/*v*)) at an injection temperature of 230 °C and analyzed in a random order. Helium was used as the carrier gas at a linear velocity of 39 cm/s. The column temperature was held at 80 °C for 2 min, increased by 15 °C/min to 330 °C, and finally kept at 330 °C for 6 min. The transfer line and ion source temperatures were 250 °C and 200 °C, respectively. Ions were generated by electron ionization (EI) at 70 eV. Mass spectra were recorded at 20 scans per second over the mass range of m/z 85–500. A standard alkane mixture (C9−C40) was injected at the beginning of the analysis to calculate the retention indices (RIs) used for tentative identification.

### 2.4. Data Processing

Chromatographic GC–MS analysis data were converted into the netCDF format using the GC–MS Solution software package (Shimadzu, Kyoto, Japan) following previously published methods [20,21]. This included file format conversion to analytical data interchange protocol (ACDF, CDF). Peak alignment detection, baseline correction, and alignments were performed using the freely available software package MetAlign and Output version 1.30, respectively. The pooled QC data were utilized in MetAlign as reference data. The processed data were then exported to the CSV-format file. Peak RIs were calculated on the basis of the retention time of the standard alkane mixture. By comparing the RIs and their mass spectra with an in-house library prepared from authentic standards, tentative identifications were performed using AIoutput2 annotation software, and the data matrix was constructed. The peaks which were not of biological origin were excluded manually from the data matrix (refer to the chromatograph of the blank). The mass spectra of all the peaks were compared with the NIST and Wiley libraries, and the retention times and the mass spectra of sugar peaks were compared with the authentic standards to confirm the tentative identifications. The assigned peak intensities were normalized against the intensity of the ribitol internal standard. 

### 2.5. Statistical Analysis

The raw chromatographic data obtained from the GC–MS analysis (*QDC file) were converted into the ANDI AIA format (*ABF file) from GCMS Solution MS data files using the GC–MS Solution software package (Shimadzu, Kyoto, Japan). Peak alignment, filtering, and annotations were completed using MS-DIAL (Riken, Tokyo, Japan). In the annotation step, the QC sample acted as a reference. The metabolites were tentatively annotated according to their RIs recorded on RI GL-Science DB (InertCap 5MS-NP, predicted Fiehn RI, 494 records), downloadable from the MS-DIAL official website. Metabolite peaks were considered if the height was five times higher than the blank (see Appendix A). Furthermore, additional filtering was applied by selecting data that showed a relative standard deviation (RSD) of less than 30% within the QC samples. Tentatively annotated metabolites were subjected to PCA using the commercial software SIMCA P+ ver. 13.0.3 package (Umetrics, Umea, Sweden) [11,21].

## 3. Results and Discussion

The soybeans used in this study were obtained from KOPTI (Organization of *Tempe* and Tofu Producers in Indonesia). Raprima (dried *Rhizopus oligosporus NRRL 2771)*, which is made by The Indonesian Institute of Sciences (LIPI–*Lembaga Ilmu Pengetahuan Indonesia*), was used as the starter inoculum for *tempe* production [22]. In general, *tempe* production is divided into two parts: (1) pre-fungal fermentation consisting of soybean soaking, outer skin dehulling, cooking of the beans, culture starter inoculation, and packing; (2) 48 h of fungal fermentation [20]. This research produced *tempe* using a method previously described by Mulyowidarso (1989) and Arbianto (1995 in Roswanjaya, 2006) with some modifications [23,24,25], as outlined in Figure 1. The modification was carried out at the boiling step where the beans were steamed rather than boiled to prevent nutritional factors especially isoflavone from leaching out from the beans into the water [26]. 

*Tempe* production in Indonesia is generally conducted for 48 h to generate a product with distinctive characteristics according to the acceptable standards of *tempe* makers (based on knowledge and experience) in Indonesia. By definition, over-fermentation occurs when the fermentation time is extended beyond 48 h [6]. During over-fermentation, the appearance of the *tempe* will change, wherein the color will turn brown, creating a distinctive texture, flavor, and odor [27]. Microorganisms, including *R. oligosporus*, yeast, and some bacteria, are crucial during this process [28]. In some studies, over-fermented *tempe* is called overripe *tempe* (*tempe semangit*). Over-fermented *tempe* can be utilized as a condiment in Javanese cuisine [29]. 

In this study, metabolite changes in each stage of *tempe* production were investigated from raw soybeans (RS), soybean soaked for 24 h (SS), steamed soybeans (StS), fungal fermentation (FF) for 48 h after inoculating the starter, and over-fermentation (OF) up to 72 h. The experiment and discussion are divided into three parts (SS, FF, OF) to address these issues. Four experiments through sets of data were used to examine the differences between the treatments a (Table 1). Soybean samples were collected every 6 h from the soaking process and every 12 h from the FF and OF processes according to the *tempe* production steps in Table 1.

### 3.1. Metabolite Changes in the SS Process

GC–MS-based metabolite profiling was carried out on aqueous soybean extracts from every stage of *tempe* production to gain a general understanding of the metabolite differences in the samples. A total of 121 metabolites were annotated from the GC–MS analysis and subjected to principal component analysis (PCA). These tentatively annotated compounds consisted of amino acids, organic acids, sugars, sugar alcohols, and other compounds. From the three treatments, differences were observed in metabolites that accumulated at different stages. 

Dataset 1 from SS in Table 1 was used to investigate metabolic transformations during the SS process and observe the differences between soybeans before and after soaking. Seven soybean samples were taken from this process. Raw soybeans (RS) refer to the beans prior to soaking in water, whereas the 0 h soaked soybeans (SS0) were the washed raw soybeans, which were taken shortly after putting into the water. SS was conducted for 24 h, and sampling was carried out every 6 h. Therefore, in 24 h of soaking, seven samples were collected. After the soaking process, the soybeans were washed, dehulled, and steamed to remove microorganisms and make them more tender and easier to begin the fermentation process. These samples were designated as StS. 

The PCA score scatterplot shows clear separation of all samples, which can be divided into two big groups as a function of principal component (PC) 1 (Figure 2). This separation shows that RS and SS0 were clustered into one group on the right side. Meanwhile, SS6, SS12, SS18, SS24, and StS were clustered together in another group on the left side. Clear separations are shown in the plot that explained 33.6% of the variability. The loading plot for PC1 showed that sugar alcohols (yellow dots) and amino acids (orange dots) were important for the separation of seven samples before and after the soaking process (Figure 2). Metabolites shown to contribute to this separation were from the sugar alcohol group on the right side of PC1 (before soaking) and the amino acid group on the other side (after soaking). At the beginning of the process, metabolites were dominated by sugar alcohols, such as mannitol, sorbitol, and propylene glycol (Figure 2B). However, sugar alcohol levels decreased as microorganisms utilized them during the soaking process. At the end of the soaking process, the metabolite profile changed as it was dominated by amino acids, namely, threonine, methionine, and lysine (Figure 2B). Threonine, lysine, and methionine are amino acids that contribute to a sweet flavor [30]. Some bacteria such as *Lactobacillus casei*, *Streptococcus faecium*, and *Staphylococcus epidermidis* dominate the soaking process and are the main species that contribute to reducing pH [25]. *Klebsiella pneumoniae*, *Klebsiella ozaenae*, *Enterobacter cloacae*, *Enterobacter agglomerans*, *Citrobacter diversus*, and *Bacillus brevis*, as well as the yeasts *Pichia burtonii*, *Candida diddensiae*, and *Rhodotorula rubra*, also made a significant contribution to the process [25]. These microorganisms caused the metabolomic changes in the soaking process by utilizing the dissolved soybean substances in the water as substrate for their growth. The metabolic end-products from growth diffused into the seed and affected its chemical composition [25].

### 3.2. Metabolite Changes in the Fungal Fermentation (FF)

Dataset 2 (Table 1) was used in the second step of the tempe production process to examine how metabolites change during FF. Five samples were collected from the tempe fermentation process at different times every 12 h. Sample FF0 was collected promptly after the dried steamed soybeans were inoculated. PCA shows that the data separation was based on the time of the fermentation process, explained with 69.9% variance (Figure 3). Figure 3A shows that samples from the first 24 h (FF0, FF12, and FF24) were in the same cluster in the negative PC1. FF0 and FF12 were clustered together in a smaller group. Results showed that there were negligible changes at the beginning of the fermentation process until after 12 h. Samples from the next 24 h (FF36 and FF48) were clustered in the positive PC1. Before fermentation, the soybeans were dominated by sugars; after fermentation, they were dominated by amino acids (glutamine) and other components, such as uracil and 2-aminoethanol.

Figure 3B shows the representative metabolites in the PC1 loading plot. Gentiobiose, galactinol, and glucarate showed higher concentrations at the beginning of the fermentation process. Conversely, the FF36 and FF48 samples had a higher concentration of uracil, glutamine, and 2-aminoethanol. The PC1 loading plot shows that sugars and amino acids were essential for separating the first and second half of the 24 h fungal fermentation process. Some sugars and sugar alcohols were found in the first 24 h (before fermentation). Streptococcus faecium, Lactobacillus casei, Klebsiella pneumoniae, Bacillus brevis, and Pichia burtonii are microorganisms known to utilize sugars such as sucrose, stachyose, raffinose, fructose, glucose, galactose, and melibiose during soybean soaking [31]. These microorganisms were also found to contribute to fungal fermentation process [14]. Hence, sugars such as gentiobiose, galactinol, and glucarate decreased toward the end of the process (Figure 3C). Amino acids increased as fermentation progressed and accumulated at the end of the process. This indicates that R. oligosporus proteolytic enzymes, which break long-chain protein molecules into shorter fragments (amino acids), were active [6,12]. Thus, uracil, glutamine, and 2-aminoethanol increased as the fermentation process proceeded. 

It can be seen that there were significant differences between the two steps of tempe production, SS (Figure 2) and FF (Figure 3). Figure 2 showed how metabolites changed in soybean soaking, in which sugar alcohols decreased and amino acids that contribute to a sweet flavor increased as soaking progressed. Figure 3 depicts the metabolites changes that occurred during the tempe fermentation by Rhizopus spp. At the end of the fermentation, a general increase was observed for amino acids, including glutamine which contributes to umami taste. In addition, uracil and 2-aminoethanol also increased after fermentation. In the subsequent analysis comparing the metabolite changes that occur throughout all the stages of tempe production process, the increase in amino acids during fermentation was much more significant compared to the increase in amino acids observed in soaking process.

### 3.3. Metabolite Changes in Tempe Production, from RS to 48 h of FF

The third experiment determined how metabolites changed throughout the whole *tempe* production process, which was monitored gradually from RS to FF (datasets 1 and 2). Figure 4A shows the score scatter plots. There was a clear separation between the two *tempe* production stages, namely, SS and FF. RS and all the samples from SS were clustered into the first group, while StS and all the FF samples were clustered together in the second group. The second group contained two small groups: the group with StS and soybeans from FF until 24 h. Another group contained soybeans from 36 h and 48 h of FF. All the treatments from RS to 24 h of SS, StS, and FF were clustered into one group on the negative side of PC1. Otherwise, FF soybeans from 36–48 h were clustered on the positive PC1 side, with 59.5% of the variability.

The PC1 loading plot shows that sugars/sugar alcohols and amino acids were essential for separating SS and FF. The distribution of the samples shows that the metabolites changed along with the fermentation time. PC1 shows that the metabolites contributing to this separation were from the sugar and sugar alcohol groups, such as sucrose, glucarate, and galactinol, while the other side included tyrosine, 3-hydroxyisovaleric acid, and homocysteine (Figure 4B,C). A previous study showed that sugars and amino acids were essential compounds separating legumes before and after fermentation [32]. 

### 3.4. Metabolite Changes in Tempe Production from RS to 72 h of OF 

Datasets 1, 2, and 3 were used to determine how the metabolites would change if FF was prolonged to 72 h (over-fermentation). Figure 5A, in general, gives a vivid distinction of the three steps of soybean treatment, namely, SS, FF, and OF, which were divided into three groups. According to PCA, all treatments from RS, SS, StS, and FF to 24 h were clustered into one group in negative PC1, explaining 59.8% of the variance. Meanwhile, FF (36 h and 48 h) and the over-fermented *tempe* (60 h and 72 h fermentation) were clustered together in positive PC1.

The distribution of the samples indicated that different soybean treatments resulted in different metabolite profiles for each fermentation time, which is consistent with the results of a previous study [32]. In this experiment, the PC1 loading plot shows that sugars or sugar alcohols, organic acids, and amino acids were essential for separating RS from FF and over-fermented *tempe*. Specifically, the metabolites that contributed to the separation were from the sugar and sugar alcohol groups, such as sucrose, glucarate, and galactinol, while the other side included tyrosine, 3-hydroxyisovaleric acid, and homocysteine (Figure 5B,C). Amino acids are compounds that accumulate in *tempe* [12]. During OF, *R. oligosporus* growth declined and was replaced by bacteria that degrade amino acids and produce compounds with a unique pungent odor [29]. 

Figure 6 compares *tempe* production (from soybean soaking to FF) and prolonged fermentation or over-fermentation (from soybean soaking to OF). The highlighted metabolites in the loading plot contributed to the separation, including sugar or sugar alcohol, organic acid, and amino acid groups. According to the loading plot of these two processes, in the beginning, sucrose, glucarate, and galactinol were the metabolites that contributed to the separation. However, in the end, only homocysteine contributed to both processes, and two other metabolites were different. At the end of FF, 4-hydroxyphenylaceticic acid and glutamine also contributed to the separation (Figure 4B,C), while tyrosine and 3-hydroxyisovaleric acid contributed to the separation of the OF process (Figure 5B,C). 

In addition to sugars, amino acids and peptides are known to contribute to the sweet taste of fermented food including *tempe* [33]. Throughout the *tempe* production process, we can see the metabolite changes that contribute to *tempe* flavor. Sugars and sugar alcohols dominated at the SS stage, whereas organic acids contributed to sour flavor, amino acids such as threonine and serine contributed to sweet flavor, and glutamine contributed to umami flavor, all of which were accumulated at the FF. At the end of OF, bitter amino acids (tryptophan, methionine, leucine, isoleucine, valine, histidine, and tyrosine) were predominant (Figure 6). 

Figure 7 shows the amino acids that contributed to the aroma and flavor of *tempe*. The bar graphs in Figure 7A depict the amino acids that contributed to the bitter taste, Figure 7B,C respectively show the amino acids that contributed to the sweet and umami flavor in *tempe* that accumulated as fermentation progressed. Similarly, amino acids related to bitter taste accumulated with a longer fermentation time. Hence, at the end of the OF, the *tempe* process produced bitter-tasting *tempe* without an interesting aroma. The dominant aromatic compound of over-fermented *tempe* was produced by bacterial proteolytic enzymes, resulting in an unappealing unique odor [34]. However, in some cities in Indonesia, it is used as a condiment in Javanese cuisine called *tempe semangit* or *tempe bosok* [29].

Some oligosaccharides in raw soybeans can cause flatulence. After fermentation of the soybeans to produce *tempe*, some oligosaccharides that contribute to flatulence decreased along with fermentation time (Figure 8A). This indicates that microorganisms might be able to metabolize sucrose; hence, it tended to decrease with fermentation time and started to disappear at FF (24 h). *Streptococcus faecium*, *Staphylococcus epidermidis*, and *P. burtonii* produce invertase to metabolize sucrose, while *K. pneumoniae* can produce both invertase and α-galactosidase [31]. Invertase and α-galactosidase hydrolyze raffinose to produce melibiose. Thus, in FF (24 h), melibiose suddenly increased and then gradually decreased as microorganisms utilized it [35]. In this study, glucose, fructose, and galactose were also detected in the process, similar to the previous report that glucose was the main substrate for fermenting microorganism [31]. Meanwhile, daidzein and genistein are isoflavone aglycons formed by microbial activity of *Bacillus cereus* detected during *tempe* production [36]. In our study, daidzein and genistein concentration continuously increased during *tempe* production and the concentration remained low at SS, but gradually increased at FF (24 h) (Figure 8B).

## 4. Conclusions

In this study, a metabolomics approach was used to determine the significant changes in the metabolite profile of each stage of *tempe* production from soybean soaking and fermentation to over-fermentation. During the whole process, sugars and/or sugar alcohols consistently dominated the metabolite profile at the beginning of every stage and decreased as fermentation progressed. At the end of the process, accumulation of amino acids was observed. Sugar alcohols (mannitol and sorbitol) dominated at the beginning of pre-fungal fermentation, i.e., the SS process, while amino acids that contribute to sweet flavor accumulated at the end of the process. Meanwhile, sugars (disaccharide and oligosaccharide) dominated the FF stage at the beginning of the *tempe* production procedure, while amino acids that contribute to umami flavor, organic acids, and isoflavones such as uracil, glutamine, and 2-aminoethanol gradually increased. When the fermentation time was extended to 72 h to produce over-fermented *tempe*, some amino acids contributing to bitter taste such as histidine, leucine, isoleucine, valine, and methionine accumulated at the end of the process. Amino acids that contribute to umami flavor started to accumulate during 24–36 h of FF and gradually increased until 72 h. Sugar flatulence components such as sucrose, raffinose, and melibiose gradually decreased during *tempe* production, while isoflavone aglycon, an antioxidant, was increased. This is the first study reporting on the metabolomics of the whole *tempe* production process. Understanding the dynamic changes in the metabolite profile at each stage of *tempe* production will be valuable to further improve the product quality of *tempe* by modulating its content.

## Figures and Tables

**Figure 1 metabolites-13-00300-f001:**
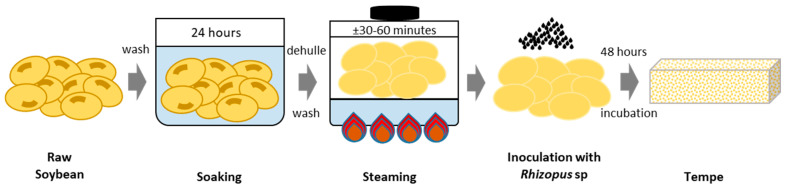
*Tempe* production process. Raw soybeans were sorted and soaked for 24 h at room temperature. They were washed and peeled to remove the skin from the bean, steamed for 30–60 min, and then cooled and dried after the steaming process was complete. Fungal fermentation was performed by pouring the starter inoculum into the dry soybeans and wrapping them in perforated plastic for 48 h. The general steps of this *tempe* production research were as follows: raw soybean (RS), soaked soybean (SS), steamed soybean (StS), fungal fermentation (FF), and over-fermentation (OF).

**Figure 2 metabolites-13-00300-f002:**
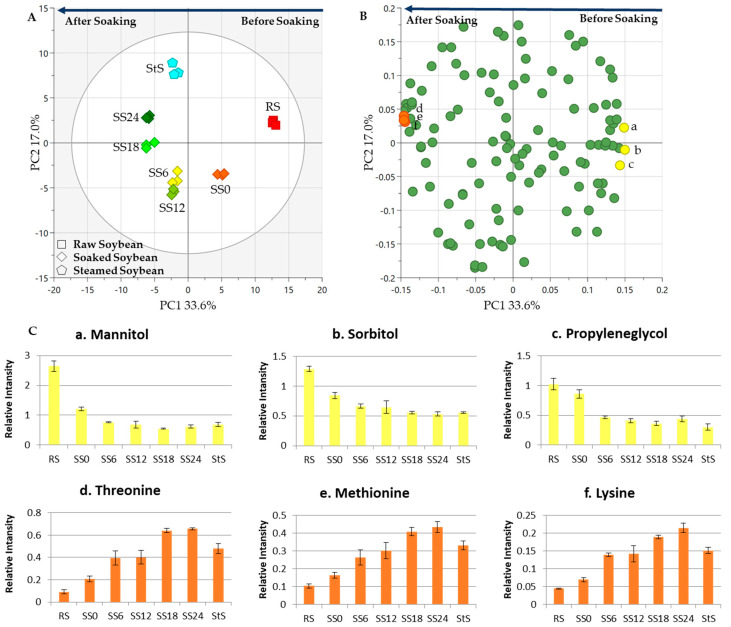
Metabolite changes during the soaking process of soybean. The PCA results of seven soybean samples were obtained to show the metabolite profiles through the soybean soaking process. (**A**) The score plot shows the separation between each sample. RS and SS0 were clustered on the positive side of principal component 1 (PC1), and the other samples clustered on the negative side. The points indicate the treatment times. Different shapes from the legend represent different treatments (RS: raw soybean, SS: soaked soybean, StS: steamed soybean), and the numbers indicate different sampling times in hours (0 h, 6 h, 12 h, 18 h, and 24 h). (**B**) Loading plot. The metabolites discussed in the main text are labeled as follows: (a) mannitol; (b) sorbitol; (c) propylene glycol; (d) threonine; (e) methionine; (f) lysine. (**C**) Bar charts. The metabolites shown in the bar chart are markers from the SS process. The vertical axis represents relative intensity. The horizontal axis represents the samples. The error bar shows the standard deviation obtained from three replicates.

**Figure 3 metabolites-13-00300-f003:**
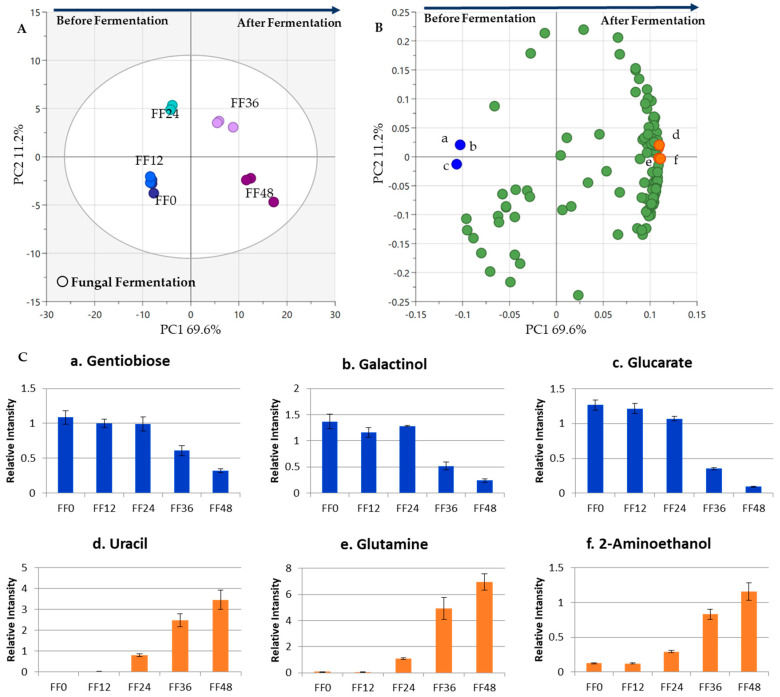
Metabolite changes during the FF process. The PCA results of five soybean samples from different sampling times were obtained to observe the metabolite profile of the fungal fermentation (FF) process. (**A**) The score plot shows the separation between each sample based on sampling time. Samples FF0, FF12, and FF24 were clustered on the negative side of PC1, while FF36 and FF48 clustered on the positive side. The points indicate the treatment times, and the numbers indicate different sampling times in hours (0 h, 12 h, 24 h, 36 h, and 48 h). (**B**) Loading plot. The metabolites as markers from the FF treatment were labeled as follows: (a) gentiobiose; (b) galactinol; (c) glucarate; (d) uracil; (e) glutamine; (f) 2-aminoethanol. (**C**) Bar charts. The metabolites shown in the bar chart are markers from FF. The vertical axis represents relative intensity. The horizontal axis represents the samples. The error bar shows the standard deviation obtained from three replicates.

**Figure 4 metabolites-13-00300-f004:**
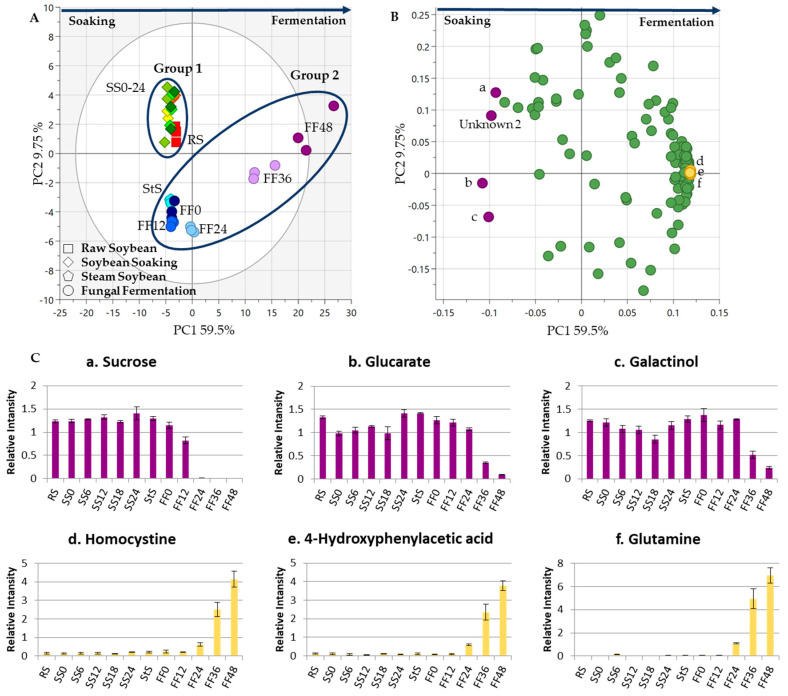
Metabolite changes during the whole *tempe* production process. The PCA results of eight soybean samples from different sampling times were obtained to observe the metabolite profiles from *tempe* production process (from RS to FF). (**A**) The score plot shows the separation between each sample based on sampling time. All the soybean soaking (SS) include RS and fungal fermentation (FF) samples were clustered separately into two groups. The points indicate the treatment times, different shapes represent different treatments (RS: raw soybeans, SS: soaked soybeans, StS: steamed soybeans, FF: fungal fermentation), and the numbers indicate the sampling times in hours (0 h, 6 h, 12 h, 18 h, 24 h, 36 h, and 48 h). (**B**) Loading plot. The metabolites are markers from the FF treatment and are labeled as follows: (a) sucrose; (b) glucarate; (c) galactinol; (d) homocysteine; (e) 4-hydroxyphenylaceticic acid; (f) glutamine. (**C**) Bar charts. The metabolites shown in the bar chart are markers from the SS and FF processes. The vertical axis represents relative intensity. The horizontal axis represents the samples. The error bar shows the standard deviation obtained from three replicates.

**Figure 5 metabolites-13-00300-f005:**
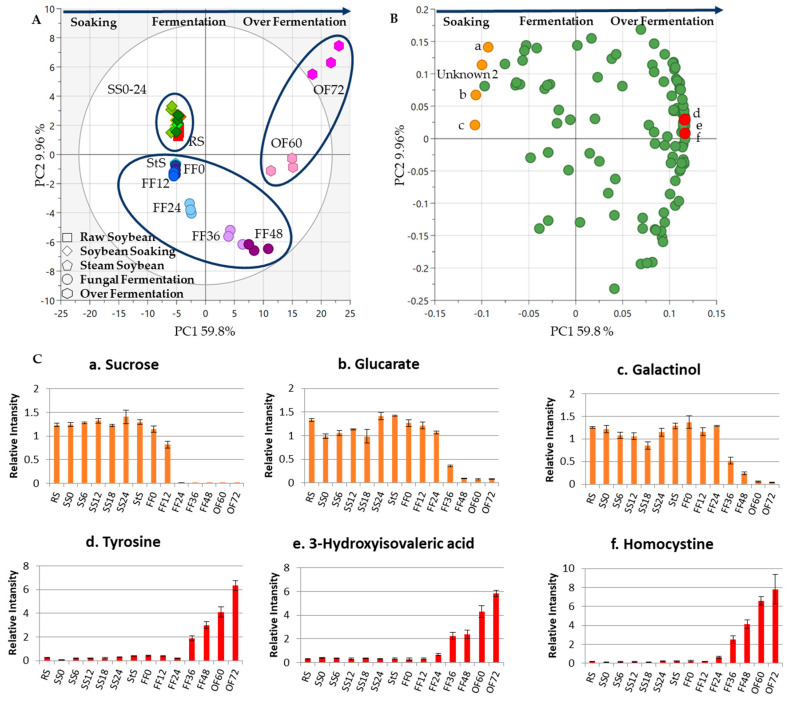
The PCA results of 14 soybean samples from different sampling times were obtained to observe the metabolite profiles from *tempe* production to over-fermentation. The metabolites changed along the process. (**A**) The score plot shows the separation between each sample based on sampling time. All soybean soaking, fungal fermentation, and over-fermented samples were clustered separately into three groups. The points indicate the treatment times, the different shapes from the legend represent different treatments (RS: raw soybean, SS: soaked soybean, StS: steamed soybean FF: fungal fermentation, OF: over-fermentation), and the numbers indicate different sampling times in hours (0 h, 6 h, 12 h, 18 h, 24 h, 36 h, 48 h, 60 h, and 72 h). (**B**) Loading plot. The metabolites are markers from the FF treatment and are lettered as follows: (a) sucrose; (b) glucarate; (c) galactinol; (d) tyrosine; (e) 3-hydroxyisovaleric acid; (f) homocysteine. (**C**) Bar charts. The metabolites shown on the bar chart are markers from the SS, FF, and OF processes. The vertical axis represents relative intensity. The horizontal axis represents the samples. The error bar shows the standard deviation obtained from three replicates.

**Figure 6 metabolites-13-00300-f006:**
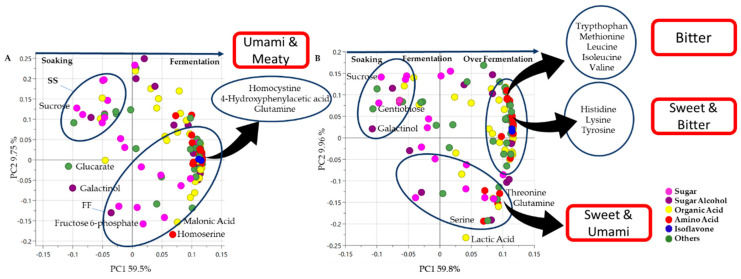
(**A**) Loading plot of soybeans during *tempe* production from soybean soaking (SS) to the 48 h fungal fermentation (FF) process. (**B**) Loading plot of soybeans from soybean soaking (SS) to 72 h over-fermented *tempe* (OF). The lilac color indicates sugars, the purple color indicates sugar alcohols, the yellow color indicates organic acids, the red color indicates amino acids, and the blue color indicates others. SS: soaked soybean, FF: fungal fermentation, OF: over-fermentation.

**Figure 7 metabolites-13-00300-f007:**
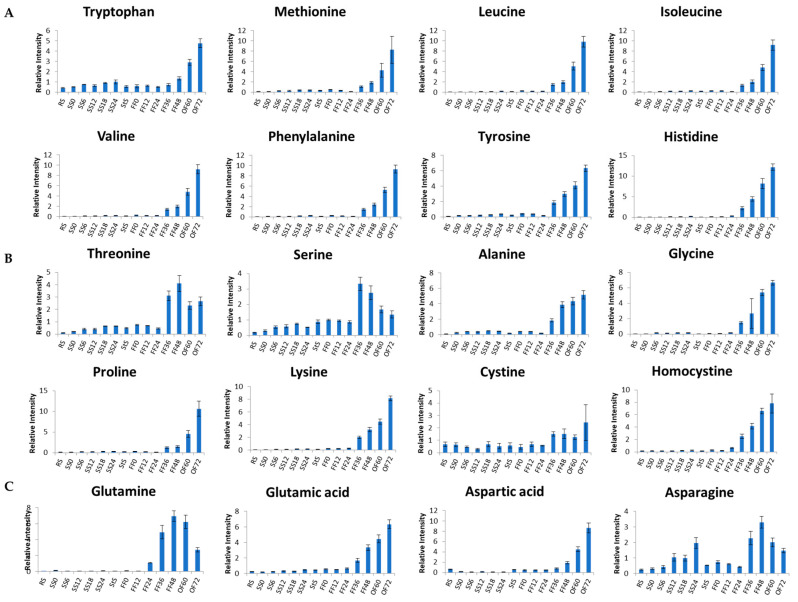
(**A**) Amino acids contributing to a bitter taste. (**B**) Amino acids contributing to a sweet flavor. (**C**) Amino acids contributing to an umami/meaty and/or sour flavor. The vertical axis represents relative intensity. The horizontal axis represents the samples. The error bars show the standard deviation obtained from three replicates.

**Figure 8 metabolites-13-00300-f008:**
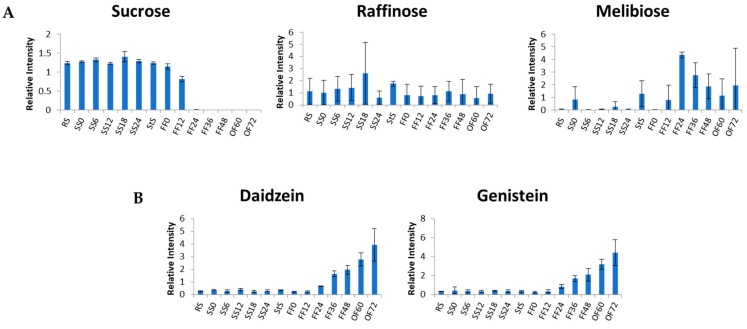
(**A**) Oligosaccharides contribute to flatulence. (**B**) Isoflavone aglycone. The vertical axis represents relative intensity. The horizontal axis represents the samples. The error bar shows the standard deviation obtained from three replicates.

**Table 1 metabolites-13-00300-t001:** Sample code list for *tempe* production.

No	Data Set	Time Treatment	Sampling Code
1.	Soybean soaking	Raw soybean	RS
		0 h	SS0
		6 h	SS6
		12 h	SS12
		18 h	SS18
		24 h	SS24
		Steamed soybean	StS
2.	Fungal fermentation	0 h	FF0
		12 h	FF12
		24 h	FF24
		36 h	FF36
		48 h	FF48
3.	Over-fermentation	60 h	OF60
		72 h	OF72

## Data Availability

No new data were created.

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
