# Peer review of "Metabolite Changes in Indonesian Tempe Production from Raw Soybeans to Over-Fermented Tempe"

_metabolites, 2023, doi:10.3390/metabo13020300_

Round 1

Reviewer 1 Report

The paper titled : “Metabolite changes in Indonesian tempe production from raw soybeans to over-fermented tempe”, submitted by the authors Prativi et al., investigated  how metabolites change overtime during the tempe production process using GC-MS that influence aroma and flavor based on the content of amino acids and sugars.

The paper contains good amount of data about the selected topic and is if special interest for researchers within this field. There are some things need to be addressed before the publishing of this paper:

1.       In the abstract

-           Please add the amount of major secondary metabolites found in FF stage because it represent the third experiment determined how metabolites changed throughout the whole tempe production process.

-          Add the prospect of the work here and in the conclusion section as well

2.       In the introduction part :

-           The introduction part is well written in my opinion

3.       In the results and discussion section,

·       Please show the significance differences on the columns of figures 2 and 3.

·       A labelling for the Y-axis in figures 2,3 (C-F) should be added.

·       Figure 7, missing the labels on the Y-axis (relative intensity) , it should be shown on the figure itself.

·       The same as above for figure 8. In addition, the figure need to be enlarged and significant differences shown on it.

·       The GC-MS chromatograms should be added in the results showing major metabolites.

4.       In the materials and methods part

-          The source of the soybean samples used and documentation should be mentioned

5.       The conclusion is missing.

I give you major revision.

Reviewer 2 Report

The manuscript by Prativi et al. describes extraction of metabolites and their GC MS profiling during the preparation of a traditional food Tempe. The authors also compared various stages of this food preparation with that of changes in the metabolites profile. The study is well planned and targeted and generally presentation style is interesting. Some minor changes can improve the manuscript.

1.     L50 Do not repeat ‘trigger’. Rephrase the sentence

2.     L63 Do not italicize ‘sp.’. Only names are italicized. Correct it throughout the paper.

3.     L74: Does it refer that all the types of amino acids are present in the fermented food? If so, rephrase the sentence.

4.     In Fig 2, in sugar profiling, what is presented in y axis, mention its units. Also discuss if the presence of sugars is corroborated with the earlier findings/reports and potential of the food for fermenting organism.

5.     In all the figures, check if y axis can be presented in a better way.

6.     Section 3.4 Avoid repetition

7.     The results and discussion section states about the fermentation of the Tempe, however, the method section does not include any information in this regard. Details are needed to better understand of the process.

Reviewer 3 Report

The manuscript deals with the investigation of the metabolite changes in Indonesian tempe production from raw soybeans to over-fermented tempe. Up to fourteen samples of raw soybeans, namely soaked soybeans, steamed soybeans, fungal fermented soybeans, and over-fermented soybeans were analyzed by gas chromatography-mass spectrometry (GC-MS) in order to get acquainted on the metabolite profiling aiming to determine soybean transformations from various fermentation times and identify disparity-related metabolites. The results achieved highlight that soybeans samples clustered together based on the different fermentation steps. Additionally, the results showed that sugar, sugar alcohol, organic acids, and amino acids as well as fermentation time contributed to the soybean metabolite profile transformations.

The work is interesting and the results achieved are of interest for the tempe industry by providing useful information to improve product quality and revise the standard of tempe production based on metabolite profiling.

Overall, the manuscript is well-written although I invite the authors to use impersonal form throughout the whole manuscript. I do have some relevant (major) remarks to be addressed prior to eventual publication.

-       Introduction. The authors must better emphasize the novelty of the work in comparison with previously published papers on the topic.

-       Lines 126-131. These lines should be moved to the 2.1 section.

-       Section 2.1. The authors are requested to include representative GC-MS chromatograms with proper peak identification (stack or super-imposed) since they add substantial value to the paper.

-  The inclusion a Table including all metabolites detected is, also, highly recommendable. Retention indices (not only experimental values but also retrieved from literature with the indication of the source) must be reported.

-  Authors are strongly encouraged to provide a true quantification at least of major components of the analyzed samples. In this context, no FID or TCD were used as detector, so in case of MS detection, data should be performed by means of reference materials.

Round 2

Reviewer 1 Report

Accepted for me 

Author Response

Dear Reviewer,

Thank you for your kind attention.

Kind Regards,

Billqys

Reviewer 3 Report

The authors have addressed partly the concerns raised in the first round of reviews.

1) I asked the authors to keep consistent and use impersonal form throoughtput the manuscript. This is still unsolve e.g. line 137: "We" carried

2)Table S2, in Table subheading there some letters a,b,c as superscript but they are not fully described at the bottom of the Table. Also, I sked to delete Retention times and use LRI (not only experimental values but also retrieved from literature with the indication of the source)

3) The authors neglected the last key remark: -  Authors are strongly encouraged to provide a true quantification at least of major components of the analyzed samples. In this context, no FID or TCD were used as detector, so in case of MS detection, data should be performed by means of reference materials.

Author Response

Dear Reviewer,

Thank you for your kind attention.

Best Regards,

Billqys

Round 3

Reviewer 3 Report

The paper has now reached final form for publication.